# Perceived Parenting Stress Is Related to Cardiac Flexibility in Mothers: Data from the NorBaby Study

**DOI:** 10.3390/bs14020117

**Published:** 2024-02-05

**Authors:** Francesca Parisi, Ragnhild Sørensen Høifødt, Agnes Bohne, Catharina Elisabeth Arfwedson Wang, Gerit Pfuhl

**Affiliations:** 1Department of Psychology, Norwegian University of Science and Technology, 7491 Trondheim, Norway; francesca.parisi@ntnu.no; 2Department of Psychology, UiT the Arctic University of Norway, 9019 Tromsø, Norway; 3Division of Child and Adolescent Health, University Hospital of Northern Norway, 9019 Tromsø, Norway

**Keywords:** heart rate variability, perinatal period, depression, parenting, infant health

## Abstract

Heart rate variability (HRV) is an indicator of autonomic nervous system activity, and high levels of stress and/or depressive symptoms may reduce HRV. Here, we assessed whether (a) parental stress affected HRV in mothers during the perinatal period and whether this is mediated by bonding and (b) whether antenatal maternal mental states, specifically repetitive negative thinking, depressive symptoms, and pregnancy-related anxiety, have an impact on infant HRV, and lastly, we investigated (c) the relationship between maternal HRV and infant HRV. Data are from the Northern Babies Longitudinal Study (NorBaby). In 111 parent–infant pairs, cardiac data were collected 6 months after birth. In the antenatal period, we used the Pregnancy-Related Anxiety Questionnaire—Revised, the Edinburgh Postnatal Depression Scale, and the Perseverative Thinking Questionnaire; in the postnatal period, we used the Parenting Stress Index and the Maternal Postnatal Attachment Scale. Higher levels of perceived parenting stress but not depressive symptoms were associated with lower HRV in mothers (τ = −0.146), and this relationship was not mediated by maternal bonding. Antenatal maternal mental states were not associated with infant HRV. There was no significant correlation between maternal HRV and infant HRV. Our observational data suggest that perceived stress reduces cardiac flexibility. Future studies should measure HRV and parenting stress repeatedly during the perinatal period.

## 1. Introduction

Parenting can be a rollercoaster, with joy and exhaustion alternating quickly. The perinatal period is a profoundly personal and subjective experience marked by a large range of emotions [1,2]. During this period, the ‘ups’ or moments of happiness and well-being can outweigh the ‘downs’ or moments of sadness and worry. Notably, most mothers perceive the first six months post birth as positive [2]. Still, this experience and the balance between ups and downs can vary greatly from person to person. Some women may feel extremely happy and excited during pregnancy and in the first months after having given birth, while others may experience periods of stress, anxiety, or depression [1,2,3,4]. Clinically significant forms of prenatal distress have reached a prevalence between 20 and 30% [5,6,7], and similarly, the prevalence of postnatal depression has reached up to 20% [8]. Notably, stress and distress may influence and be influenced by cardiac flexibility and heart rate variability (HRV), respectively. Parental stress during the perinatal period might not be an exception.

### 1.1. Heart Rate Variability

Heart rate variability (HRV) is the rate of variation between each heartbeat over time. It is an indicator of the activities of the autonomic nervous system [9], with higher HRV indicating greater cardiac autonomic flexibility and lower HRV indicating greater rigidity and loss of autonomic nervous system control over the circulatory system [10]. Reduced HRV is linked to both somatic diseases, e.g., congestive heart failure [11], sleep disturbance [12], and increased inflammation [13], and mental disorders, e.g., stress [14], anxiety [15], and depression [16]. An association between depression and reduced HRV has been reported in several studies [17,18,19,20,21], although the precise relationship between depression and HRV is not known. HRV is linked to several important functional domains [19], e.g., increased HRV has been associated with better emotional regulation and stress resilience [10,22,23,24], and high HRV is generally regarded as a marker of superior self-control and environmental adaptation [25]. Conversely, chronic stress leads to a more dominant sympathetic system and a decrease in HRV, also known as the “fight-or-flight” response [14,26]. An alternative to the “fight-or-flight” response is “tend-and-befriend” behavior [27], particularly in the nurturing context. “Tend-and-befriend” is a behavior aimed at protecting one’s offspring and seeking mutual defense by being part of, building, and maintaining a social group. This “tend-and-befriend” response might be mediated by oxytocin [28], a hormone released during bonding and shown in men to increase HRV [18]. Thus, receiving social support may act as a buffer against stress and contribute to a well-functioning autonomous system.

In sum, high HRV is associated with a better ability to cope with stressors. Previous studies have mainly investigated daily, chronic, and total life stress [29,30], whereas only a few studies have investigated parenting stress and HRV [31,32]. HRV is negatively correlated with stress [33,34,35], which leaves open whether stress affects HRV or whether HRV affects stress perception and regulation.

### 1.2. Parental Stress and Well-Being

Parental characteristics like emotional well-being, birth expectations, and past traumatic experiences [36] in conjunction with the characteristics of the child, as well as social support, contribute to shaping individual differences in parenting. These determinants of parenting interplay [36], e.g., mothers with high levels of distress were more inclined to perceive an infant’s temperament as more difficult [37] and, conversely, infant temperament contributed to parenting stress [38]. Furthermore, factors such as being a single parent, low socioeconomic status, and household chaos can lead to (additional) maternal sleep deficits [39,40], which has a bi-directional relationship with perceived stress [41]. Thus, there are adverse factors in the perinatal period that may increase perceived stress and reduce the emotional well-being of the parent. 

In the antenatal period, birth expectations and previous traumatic experiences can negatively impact pregnancy. In the postnatal period, expectations, perceived stress, the behavior of the infant, and not least how family and peers support one can impact the first months after childbirth [36]. Parents must address specific challenges related to infant care, which can be perceived as highly stressful [26]. Moreover, high parental stress can lead to depression in mothers [42], and postpartum depression or anxiety [43,44] can further increase perceived parenting stress. Parity does not appear to play a relevant role in parenting stress [45] but seems to affect postpartum depression [46]. Indeed, nulliparous women have a higher risk of postpartum depression compared to multiparous women [46]. Mothers with postpartum depression may be vulnerable to experiencing more parenting stress given that they may engage in excessive and repeated negative thinking [47,48,49]. This may also reduce new mothers’ “tend-and-befriend” behavior, which may contribute to establishing a strong bond with their newborn. Indeed, mothers who report a better bond with their newborn also report lower levels of parenting stress [50,51]. Thus, our first research question investigated whether perceived parenting stress and depressive symptoms are associated with maternal HRV, with maternal bonding mediating the relationship between stress and HRV.

### 1.3. Maternal Mental States and Infant Heart Rate Variability

Persons with elevated stress, anxiety, and depressive symptoms tend to have reduced parasympathetic function [20,29,30], which is measurable as lower HRV. If a pregnant woman experiences elevated anxiety and distress, it might affect the fetus and then the newborn [15,52]. A range of studies have found that infants’ HRV is influenced by antenatal maternal psychological states, such as stress, anxiety, and depression [14,53,54,55]. However, it is challenging to determine which of the variables (e.g., stress, anxiety, and depression) has a primary role and how long-lasting the influence is. 

Maternal anxiety during pregnancy [53,54] and a history of previous anxiety disorders [15,56] are inversely related to children’s HRV. However, only a few studies [53] have focused on antenatal-specific anxiety, which can be distinguished from general anxiety [57]. Moreover, antenatal-specific anxiety is a greater predictor of unfavorable child outcomes than general anxiety [58].

Concerning stress, Jacob et al. [55] discovered a negative association between antenatal maternal stress and infants’ HRV. However, Eykens et al. [54] identified no correlation. The disparity can be explained by the methodological procedures used. Jacob et al. [55] relied on objective life stressor detection, whereas Eykens et al. [54] relied on self-reported stress.

Regarding depression, some research studies [55,59,60] have revealed that there is no relationship between maternal depression and infants’ autonomous nervous system (ANS) functioning, whereas other studies [61] have discovered lower vagal modulation, yielding lower HRV [62], in the infants of depressed mothers. In our second research question, we investigated whether adverse antenatal factors, e.g., pregnancy-related anxiety, depressive symptoms, and recurrent negative thoughts, are associated with the infant’s HRV six months after birth.

### 1.4. Maternal and Infant Heart Rate Variability

Van den Bergh et al. [15] found an association between the HRV of infants and mothers with a history of anxiety disorders. It is not known whether this is unique to anxiety disorders or whether it would also apply to depression or more generally to mothers who have recurrent negative thoughts. Hence, we investigated whether there is a relationship between the mother’s HRV and the infant’s HRV.

### 1.5. Study Aims

Our three research questions were:


*Research question 1: Are parenting stress or depressive symptoms related to cardiac autonomic flexibility? Would maternal bonding mediate the relationship?*



*We hypothesized that the higher the perceived parenting stress and/or postnatal depressive symptoms the lower the HRV in mothers. Maternal bonding might mediate the relationship between parenting stress/depressive symptoms and HRV.*



*Research question 2: Do antenatal factors affect the infant’s cardiac autonomic flexibility? We investigated if pregnancy-related anxiety, antenatal depressive symptoms, and recurrent negative thoughts during pregnancy have an effect on the 6-month-old infant’s HRV. We hypothesized that the higher the load of maternal adverse factors during pregnancy, the lower the infant’s HRV.*



*Research question 3: Is there a relationship between the mother and infant’s cardiac autonomic flexibility? To address this, we correlated maternal and infant HRV.*


## 2. Materials and Methods

### 2.1. Participants and Procedures

The current quantitative and observational study is part of the Northern Babies Longitudinal Study (NorBaby) [63], which looks at risk and protective factors for parental mental health, parent–infant interaction, and infant development in a non-clinical sample. The research comprised 220 women (approximately 12% of pregnant women in the region) and 130 partners from the municipality of Tromsø. All Norwegian-speaking pregnant women and their partners were eligible for inclusion in the study. During their pregnancy, they were recruited by midwives and then contacted by a research team member to schedule a meeting for enrolment in the study. The data were collected between October 2015 and December 2017, and each participant completed six measurement points: T1–T3 during pregnancy (T1: 16–22 weeks gestation; T2: 24–30 weeks gestation; T3: about 31 weeks gestation) and T4–T6 after birth (T4: 6 weeks postpartum; T5: four months postpartum; T6: six months postpartum). Online questionnaires, computerized cognitive tests, videotaped observations of mother–infant interactions, and Bayley’s Scales of Infant and Toddler Development [64], a standardized evaluation of the child’s cognitive, linguistic, and motor development, were used to gather the data. Data were collected during a meeting with a member of the study team at the university (T1, T5, and T6) or at home (T2 and T3). Questionnaire data were completed at home for all time points, except for T1.

Heart rate was measured at T6, and 111 parents (106 mothers and 5 fathers) and their infants completed this measurement point.

### 2.2. Measures

#### 2.2.1. Demographic and Health Information

Demographic and health information was collected at T1. Demographic information included maternal age, education, gross yearly family income, marital status, whether the pregnancy was desired, and if this was the parents’ first child. The participants were also asked questions about their previous mental health [63]. There were two yes/no questions regarding social support—about family/friends that can help if needed; hence, the sum score for social support ranged from 0 to 2.

#### 2.2.2. Pregnancy-Related Anxiety

The Pregnancy-Related Anxiety Questionnaire—Revised [57] was used to assess pregnancy-related anxiety. Ten items assess fear of giving birth, concerns about having a physically or mentally disabled child, and concerns about one’s own appearance. The response options range from 1 (definitely not true) to 5 (definitely true), with total scores ranging from 10–50. Higher scores indicate more pregnancy-related anxiety. The internal consistency was good; McDonald’s ω = 0.856.

#### 2.2.3. Depressive Symptoms

Depressive symptoms were collected at T1 and T5 and were assessed using the Edinburgh Postnatal Depression Scale [65], a 10-item self-report validated questionnaire. It is a screening test used to detect major depression and/or anxiety during pregnancy or in the postpartum period. Items are rated from 0 to 3, with a sum score ranging from 0–30. Items 3, 5 to 10 are reverse scored. More depressive symptoms are indicated by higher scores. A score of ≥10 is suggested as a cut-off score for possible clinical depression in a study based on a Norwegian sample [66]. The internal consistency was acceptable and good; at T1, McDonald’s ω = 0.782, and at T5, McDonald’s ω = 0.837.

#### 2.2.4. Repetitive Negative Thinking

Repetitive negative thinking (RNT) was measured at T1 and T4 using the Perseverative Thinking Questionnaire [67]. The PTQ is a transdiagnostic questionnaire that assesses whether thoughts are repetitive, intrusive, and difficult to disengage from, as well as whether they are perceived as unproductive, and captures mental capacity. The questionnaire contains 15 statements about one’s thoughts. The response options range from 0 (never) to 4 (almost always), with total scores ranging from 0–60. Higher scores indicate a higher level of repetitive negative thinking. The internal consistency was excellent; at T1, McDonald’s ω = 0.944, and at T4, McDonald’s ω = 0.942.

#### 2.2.5. Parenting Stress

The Parenting Stress Index [68] was assessed at T5, and it measures the level of stress parents experience in their parental role. It contains 120 items. The index is divided into three sections: the parent domain (PD), the child domain (CD), and the life stress scale (LSS). At T5, the parent and child domains were evaluated, but the life stress domain was not assessed. The parent domain contains 54 items and assesses the stress associated with being a parent. It evaluates several areas, e.g., competence, isolation, and attachment. The child domain includes 47 items assessing stress related to child behavior, e.g., distractibility/hyperactivity, adaptability, and demandingness. The items are graded using a five-point Likert scale. The overall range is 54–270 for PD and 47–235 for CD. Higher stress levels are indicated by higher scores. The internal consistency ranged from excellent to good; the parent domain McDonald’s ω = 0.923; the child domain McDonald’s ω = 0.893.

#### 2.2.6. Postnatal Bonding

Mother-to-infant bonding was assessed at T5 using the Maternal Postnatal Attachment Scale [69]. It is a self-report questionnaire consisting of 19 items. It evaluates four ‘indicators’ of mother-to-infant bonding: proximity pleasure, tolerance, need gratification and protection, and knowledge acquisition. Items are rated on 2-, 3-, 4-, or 5-point scales, and all items are re-coded to scores ranging from 1 (poor bonding) to 5 (strong bonding) to ensure equal weighting. The overall range is from 19 to 95. Internal consistency was good; McDonald’s ω = 0.826.

#### 2.2.7. Cardiac Data

Cardiac data acquisition was carried out at T6 for both parents and infants. The measurement was completed while the babies were placed on their parents’ laps, by a table, facing a person from the research group performing the cognitive and language subscales of the Bayley Scales of Infant and Toddler Development Screening [64], a neuropsychological test that measures cognitive, language, and motor development. This took approximately 15–20 min to complete.

Two cardiac electrodes were placed axially on the left and right rib cages, with a third placed on the sternum (ground). The acquisition sample rate was set to 1000 Hz.

BioPac Acqknowledge software (version 4) was used to preprocess the electrocardiogram (ECG) records. The data were transformed with the template correlation function. The R-wave times of heartbeats were automatically detected, and then, records were visually inspected. Actual peaks that were not automatically detected were added, misidentified peaks were deleted, and noise was removed. We decided not to perform missing data estimation and instead based our analysis on only actual R-wave times to avoid any introduced effects from the estimated data [70]. We calculated the root mean square of successive interbeat interval differences (RMSDD) as the RMSSD is the primary time-domain measure for vagally mediated changes reflected in HRV [71]. Beats per minute (BPM) were calculated too.

### 2.3. Data Analysis

R software (version 4.1.2, R Teams, 2022) was used for the statistical analyses.

For our first research question, five dyads were excluded as the HRV was from the fathers. At T5, ten participants who took part at T6 did not take part, one participant had no EPDS data, and three more participants had no maternal bonding data. Furthermore, we excluded one participant due to the HRV being more than five SDs above the mean HRV of the sample. Thus, the final sample size was N = 96 for the bivariate correlation between parental stress and HRV, N = 95 for depressive symptoms and HRV, and N = 93 when mediating with maternal bonding.

Given our small sample size [72] and the interrelatedness of the variables, we used bivariate rank correlations using Kendall’s τ. For research question 1, this included: the parental stress index—parent domain (PSI-PD), the parental stress index—child domain (PSI-CD), the Edinburgh postnatal depression scale (EPDS), and maternal bonding (MPAS), all self-rated at T5, social support (from T1), and heart rate variability (RMSSD) measured at T6. Since we found no difference by parity for any of the T5 measures, we did not control for parity (see https://osf.io/na49r/, accessed on 17 November 2023). 

We performed a mediation analysis with HRV as the outcome, parental stress and depressive symptoms as predictors, and maternal bonding as the mediator.

For the second research question, we used the following T1 data from mothers: pregnancy-related anxiety, maternal depressive symptoms, and repetitive negative thinking. Two infant HRV values were outliers with more than three SDs from the mean HRV and were excluded, i.e., *n* = 109 (the results do not change when including the two values; see https://osf.io/na49r/, accessed on 17 November 2023). Given the small sample size and explorative nature, we again investigated the bivariate rank correlations.

For the third research question, we used maternal and infant cardiac data at T6 and used rank correlation to investigate the association between mothers’ and infants’ HRV.

## 3. Results

Our sample was a low-risk sample, with few reporting depressive symptoms. Table 1 provides an overview of the demographic variables at T1, the predictor variables measured at T1 and T5, and the heart rate data measured at T6.

### 3.1. Research Question 1: Are Perceived Parenting Stress and/or Depressive Symptoms Related to Maternal HRV? If So, Does Maternal Bonding Mediate the Relationship?

As seen from Table 1, few participants reported depressive symptoms at T5, i.e., the median EPDS was three, and out of the sample, only six (6.3%) mothers scored over the cut-off point of ten. The parent and child domains of the PSI were highly positively correlated (τ = 0.455, *p* < 0.001), and the parent and the child domains of the PSI were positively correlated with depressive symptoms. The more social support, the lower the perceived parenting stress in the parent domain (τ = −0.1, *p* = 0.236). The stronger the bonding, the lower the parenting stress in both domains, as well as there being fewer depressive symptoms (*p*’s < 0.001). Stronger bonding was non-significantly associated with higher HRV (τ = 0.012, *p* = 0.863). Parenting stress was significantly associated with HRV, i.e., the more perceived stress in the child domain, the lower the HRV (τ = −0.146, *p* = 0.037), and the more perceived stress in the parent domain, the lower the HRV (τ = −0.131, *p* = 0.061) too.

Given the results from the bivariate rank correlations (Figure 1), we performed a mediation analysis with PSI-PD, PSI-CD, and EPDS as predictors, MPAS as a mediator, and HRV as the outcome. Mediation analysis revealed a significant total effect of the parental stress—parent domain on HRV (*p* = 0.036, β = −0.006), with a significant direct path (*p* = 0.005, β = −0.008), but the indirect path was not significant (*p* = 0.054, β = 0.002). Thus, more perceived parenting stress was associated with lower HRV, and this relationship was not mediated by bonding. The direct and total effects of the parenting stress—child domain and depressive symptoms were not significant (see https://osf.io/na49r/, accessed on 17 November 2023). 

Thus, the bivariate correlation yielded a significant association between the parenting stress—child domain and HRV, but when controlling for bonding, it was the parent domain that had a significant association with HRV. We therefore ran a final mediation analysis using the total parenting stress score. Here, the direct effect was significant (β = −0.024, *p* = 0.013), the indirect effect was not significant (β = 0.012, *p* = 0.081), and the total effect was significant (β = −0.011, *p* = 0.047). The path from bonding to HRV was not significant (β = −0.006, *p* = 0.067). Thus, higher (overall) perceived parenting stress was associated with lower HRV, and this was not mediated by bonding.

Regarding our first research question, we found some support for higher perceived parenting stress being associated with lower cardiac autonomic flexibility. Of note, the shared variance is less than 2%.

### 3.2. Research Question 2: Are Antenatal Factors Related to the HRV of the Infant?

Antenatal factors such as anxiety or repetitive negative thinking may transmit to the fetus and influence cardiac autonomic flexibility in the infant. We therefore explored whether antenatal maternal mental states like repetitive negative thinking, depressive symptoms, and pregnancy-related anxiety were related to the infant’s HRV at T6 (six months). Anxiety, depressive symptoms, and negative thoughts were higher among nulliparous women compared to multiparous women (see https://osf.io/na49r/, accessed on 17 November 2023). Figure 2 displays the bivariate rank correlations among the seven variables. There was no relationship between depressive symptoms and the HRV of the infant, τ(109) = 0.116, *p* = 0.086, or between the infant’s HRV and pregnancy-related anxiety or repetitive negative thinking (both *p*-values > 0.1). Depressive symptoms were positively associated with repetitive negative thinking (*p* = 0.002) and anxiety (*p*-values < 0.001). Conversely, repetitive negative thinking was positively associated with anxiety (fear of birth: *p* = 0.007, fear for child: *p* = 0.042, and looks: *p* > 0.001). 

Regarding our second research question, we found no support for antenatal factors influencing the infants’ heart rate variability at six months. 

### 3.3. Research Question 3: Are Maternal and Infant HRV Associated?

There was a positive but not statistically significant correlation between maternal and infant HRV (τ(104) = 0.102, *p* = 0.102, 95% CI [−0.027; 0.244]). The results are similar when including the outliers (τ(106) = 0.104, *p* = 0.115, 95% CI [−0.033, 0.24]). Regarding our third research question, there might be a relationship between maternal and infant heart rate variability given the positive effect size, although the relationship was non-significant.

## 4. Discussion

The present study explored the relationship between maternal postnatal psychological wellness and maternal HRV, maternal antenatal psychological wellness and her infant’s HRV, and the relationship between maternal and infant HRV at six months in a low-risk, well-educated sample.

Regarding the first research question, our findings are in line with previous research associating higher stress with lower HRV [23,24,29,30]. On the one hand, parenting stress is the major stressor among Norwegian mothers, as the welfare system provides financial support and job loss is also very unlikely. On the other hand, parenting stress is a mix of psychological stress (e.g., own expectations), psychosocial stress (e.g., coping with the expectations of the partner and the infant), and physical stress (sleep deprivation and bodily changes). Our study did not distinguish between them. 

Given that only six mothers reported depressive symptoms above the cut-off point, it is not surprising to find no relationship between postnatal depressive symptoms and HRV (but see [17,21]). Our findings do not support a link between maternal bonding and HRV, a link not previously investigated. Note that bonding was high, and only very few mothers reported low bonding with their infant. It is therefore worthwhile to investigate the association between bonding and cardiac flexibility in a larger and more heterogeneous sample with respect to bonding.

Concerning the second research question, our data do not support the existence of a relationship between antenatal maternal mental states and infant HRV. In particular, in contrast with previous research [15,53,54,56], our results do not support an influence of maternal pregnancy-related anxiety on the infant’s HRV. However, most previous studies have considered general anxiety [57], which is different from pregnancy-related anxiety. The only study that has explored pregnancy-related anxiety, to the best of our knowledge, found that antenatal maternal PRAQ anxiety was associated with increased HRV parameters in the infant, which is at odds with theories predicting lower HRV in anxiety [53]. Regarding depressive symptoms, our results are in line with those of a recent study that also found no association [55,60,61]. Repetitive negative thinking (ruminations) had no relationship with the infant’s HRV in the present study. To the best of our knowledge, repetitive negative thinking has only been studied in relation to psychological states [73] and parent–infant bonding [47], but given the perseverative cognition hypothesis [74] and the role rumination plays in the genesis, maintenance, and recurrence of depression and stress [75,76,77], one could expect that repetitive negative thoughts reduce cardiac flexibility. It is also a significant predictor of depressive symptoms in the current sample [78].

Finally, we found a non-significant positive relationship between maternal and infant HRV with a small effect size. Larger samples may make this finding statistically significant but may not change the effect size. Given that many factors influence HRV, the small effect size is not surprising. 

### 4.1. Strengths and Limitations

An obvious limitation is that the NorBaby study is an observational study, and no causal inference can be made. Furthermore, our sample was well functioning, with low levels of depressive symptoms, high education levels, and high scores for mother–infant bonding. HRV was measured around the age of six months during the Bayley assessment. Not all parents took part, diminishing our sample size and power. A strength was the longitudinal design combining antenatal and postnatal self-reported measures with physiological assessment. We also recorded HR from the caregiver and the infant for over 20 min, allowing reliable estimates of HRV. By using a Norwegian sample, socioeconomic worries during the perinatal period are minimal, a confounder to well-being in many other countries.

### 4.2. Conclusions

In this low-risk sample, there was a relationship between mothers’ perceived parenting stress and maternal HRV. There was no influence of maternal antenatal factors like pregnancy-related anxiety, depressive symptoms, or repetitive negative thinking on the infant’s HRV. There was no significant association between the mother and infant’s HRV.

## Figures and Tables

**Figure 1 behavsci-14-00117-f001:**
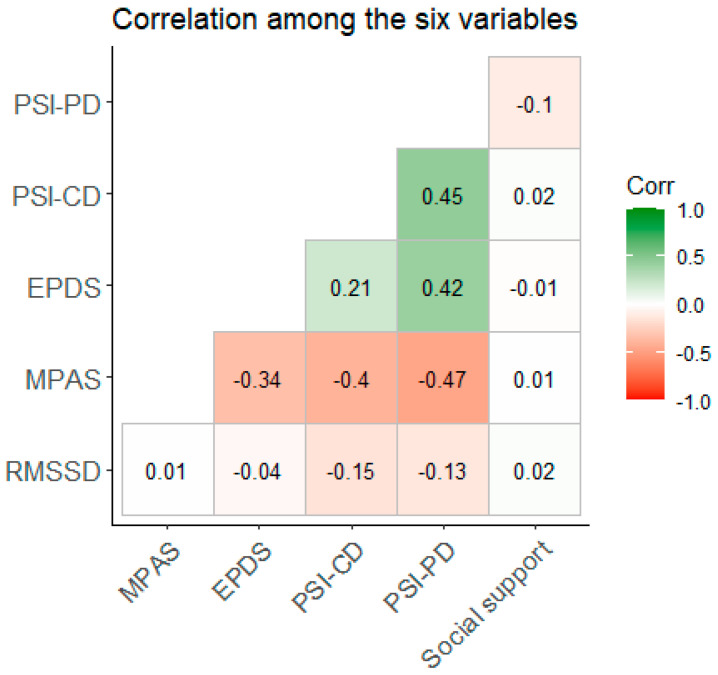
Bivariate rank correlation (Kendall’s τ) among the six variables. Legend: PSI-PD: Parent Stress Index-Parent Domain; PSI-CD: Parent Stress Index-Child Domain; EPDS: Edinburgh Postnatal Depression Scale; MPAS: Maternal Postnatal Attachment Scale; RMSSD: root-mean-square differences of successive R–R intervals (heart rate variability).

**Figure 2 behavsci-14-00117-f002:**
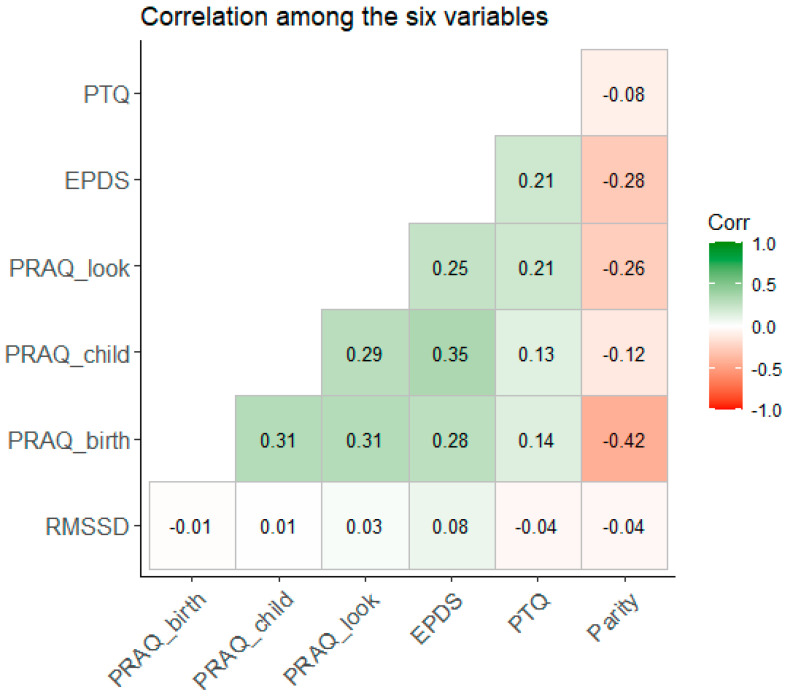
Bivariate rank correlation (Kendall’s τ) among the seven variables. Legend: PTQ: Perseverative Thinking Questionnaire; EPDS: Edinburgh Postnatal Depression Scale; PRAQ_look: subscale “Concern about own appearance” of the Pregnancy-Related Anxiety Questionnaire—Revised; PRAQ_child: subscale “Worries about bearing a physically or mentally handicapped child” of the Pregnancy-Related Anxiety Questionnaire—Revised; PRAQ_birth: subscale “Fear of giving birth” of the Pregnancy-Related Anxiety Questionnaire—Revised; RMSSD: root-mean-square differences of successive R–R intervals (heart rate variability).

**Table 1 behavsci-14-00117-t001:** Descriptive statistics.

Variables (Timepoint)	n	Mean	SD	Min	Max	Median
Age	111	30.93	4.01	20	40	31
Marital status married/cohabiting /other	111 32/78/1					
Parity nulliparous/multiparous	111 58/53					
Education high school/BA/MA or higher	111 13/35/63					
Social support	96	1.85	0.38	0	2	2
PRAQ—fear of birth	111	6.92	3.17	3	15	7.0
PRAQ—fear for child	111	10.89	3.98	4	20	11
PRAQ—looks	111	6.1	3.37	3	15	5
PTQ	111	17.72	9.60	0	47	17
EPDS (T1)	111	4.55	3.32	0	16	4
EPDS (T5)	96	3.65	3.39	0	13	3
PSI-PD (T5)	96	2.17	0.41	1.41	3.5	2.15
PSI-CD (T5)	96	1.84	0.33	1.21	3.26	1.83
MPAS (T5)	94	4.34	0.43	2.23	4.95	4.44
HRV mother (sec)	107	0.04	0.03	0.01	0.24	0.03
HRV infant (sec)	109	0.011	0.004	0.004	0.024	0.011

Legend: PRAQ—fear of birth: subscale “Fear of giving birth” of the Pregnancy-Related Anxiety Questionnaire—Revised; PRAQ—fear for child: subscale “Worries about bearing a physically or mentally handicapped child” of the Pregnancy-Related Anxiety Questionnaire—Revised; PRAQ—looks: subscale “Concern about own appearance” of the Pregnancy-Related Anxiety Questionnaire—Revised; PTQ: Perseverative Thinking Questionnaire; EPDS (T1): Edinburgh Postnatal Depression Scale measured at time 1; EPDS (T5): Edinburgh Postnatal Depression Scale measured at time 5; PSI-CD (T5): Parent Stress Index—Child Domain measured at time 5; PSI-PD (T5): Parent Stress Index—Parent Domain measured at time 5; MPAS (T5): Maternal Postnatal Attachment Scale measured at time 5; HRV mother: heart rate variability of the mother; HRV infant: heart rate variability of the infant.

## Data Availability

Data are available at https://osf.io/na49r/ (accessed on 17 November 2023).

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
