# Peer review of "Perceived Parenting Stress Is Related to Cardiac Flexibility in Mothers: Data from the NorBaby Study"

_behavsci, 2024, doi:10.3390/bs14020117_

Round 1

Reviewer 1 Report

Comments and Suggestions for Authors

Thank you for the opportunity to review this manuscript. It is well structured and the various elements are described clearly.

May I just suggest to alter the sentences starting with a number:

e.g. p4 line 172 - instead of starting the sentence with the numerical '10....' I suggest to alter this to : " Ten....'

Please adjust wherever else applicable.

The EDS is a validated tool to assess depression and/or anxiety (not depression alone) experienced within the 7 days prior to assessment (hence has its limitations). I would add that it is a 'validated' tool as this may assist the unseasoned reader to the credibility of this tool.

It seems to be assumed that the reader knows this is a quantitative study. Can this please be clearly stated somewhere relevant- e.g. Method/ Data Analysis

Table 1  - Can you label it more simply as demographic data (rather than ....RQ 1). Can there be horizontal lines separating the various values so the table is more easily readable. The legend also has some abbreviations missing such as HRV - please identify ALL abbreviations. Each table needs to stand on its own. Please apply to all tables.

P7 - The research question mentioned here - can you please align with the three mentioned AIMS rather than introducing research questions here?

Where applicable, please add a 0 in front of the decimal point rather than presenting the values as .002 present them as 0.002. It improves readability.

Figures need to be labelled underneath the figure.

Please ensure that numbers within a sentence that are under 10 are written out in full- e.g. p 10 line 338 (write six instead of 6 here).

All the best with the publication of this paper.

Author Response

Thank you very much for your constructive review and encouraging words.

We now state explicitly that it is a quantitative, observational study in the method section. 

We have revised Table 1 and provide all abbreviations, thank you.

We have rewritten the section on study aiums, summarising the research questions and our operationalisation of them.

We have added a 0 in front of all statistical values and written out numbers below 10.

Reviewer 2 Report

Comments and Suggestions for Authors

The authors tried to clarify some interesting questions about the relationship between maternal postnatal psychological wellness and her own HRV, maternal antenatal psychological wellness and her infant’s HRV, and maternal and infant HRV in a well-selected sample. The small sample requires later studies and analyses. However is a very acceptable effort, which deserves a congratulation.

I suggest reviewing Table 1, Please reduce interline space. Also, move the social support variable after the Education variable.

Author Response

Thank you very much for your kind words. We agree that larger sample size is desirable. However, as many before us, this is a time and cost issue. We provide our data, and hope that future meta-analyses will use them.

Thank you for the suggestions regarding Table 1. We have modified the table, reducing spacing and re-ordering the variables.

Reviewer 3 Report

Comments and Suggestions for Authors

The manuscript investigates heart rate variability (HRV) in mothers in the perinatal period and the association between psychosocial stress, depressive symptoms and bonding and the relationship between maternal and infant HRV. The authors report that higher parenting stress was significantly associated with lower HRV, but that was not significant for depressive symptoms. Antenatal risk factors from the mother's side were not associated with altered infant HRV. 

The study topic is of interest, the manuscript in general well written and the methodology seems sound. However, I have some questions and suggestion,

1. abstract: the following sentence is unclear: "(...) , and pregnancy-related anxiety affect infant HRV (...)"

2. Figure 1and 2: please add a legend about what is shown there and explain abbreviations

3. I would not discuss results that are not statistically significant, given also the relatively small sample size.

Author Response

Thank you very much for your constructive review.

We have rewritten the sentence in the abstract.

We provide figure legends and abbreviations. 

We only briefly discuss our non-significant results, please do note that the effect size is small but that this is as to be expected. A larger sample may yield a p- value below 0.05 but may not change the effect size.